# Safety and Immunogenicity of Recombinant *Bacille Calmette-Guérin* Strain VPM1002 and Its Derivatives in a Goat Model

**DOI:** 10.3390/ijms24065509

**Published:** 2023-03-14

**Authors:** Julia Figl, Heike Köhler, Nadine Wedlich, Elisabeth M. Liebler-Tenorio, Leander Grode, Gerald Parzmair, Gopinath Krishnamoorthy, Natalie E. Nieuwenhuizen, Stefan H. E. Kaufmann, Christian Menge

**Affiliations:** 1Institute of Molecular Pathogenesis, Friedrich-Loeffler-Institut, 07743 Jena, Germany; 2Vakzine Projekt Management GmbH, 30625 Hannover, Germany; 3Department of Immunology, Max Planck Institute for Infection Biology, 10117 Berlin, Germany; 4Max Planck Institute for Multidisciplinary Sciences, 37077 Göttingen, Germany; 5Hagler Institute for Advanced Study, Texas A&M University, College Station, TX 7843, USA

**Keywords:** VPM1002, *Bacille Calmette-Guérin*, immune response, tuberculosis vaccine, goat model

## Abstract

A more effective vaccine against tuberculosis than *Bacille Calmette-Guérin* (BCG) is urgently needed. BCG derived recombinant VPM1002 has been found to be more efficacious and safer than the parental strain in mice models. Newer candidates, such as VPM1002 Δ*pdx1* (PDX) and VPM1002 Δ*nuoG* (NUOG), were generated to further improve the safety profile or efficacy of the vaccine. Herein, we assessed the safety and immunogenicity of VPM1002 and its derivatives, PDX and NUOG, in juvenile goats. Vaccination did not affect the goats’ health in regards to clinical/hematological features. However, all three tested vaccine candidates and BCG induced granulomas at the site of injection, with some of the nodules developing ulcerations approximately one month post-vaccination. Viable vaccine strains were cultured from the injection site wounds in a few NUOG- and PDX- vaccinated animals. At necropsy (127 days post-vaccination), BCG, VPM1002, and NUOG, but not PDX, still persisted at the injection granulomas. All strains, apart from NUOG, induced granuloma formation only in the lymph nodes draining the injection site. In one animal, the administered BCG strain was recovered from the mediastinal lymph nodes. Interferon gamma (IFN-γ) release assay showed that VPM1002 and NUOG induced a strong antigen-specific response comparable to that elicited by BCG, while the response to PDX was delayed. Flow cytometry analysis of IFN-γ production by CD4^+^, CD8^+^, and γδ T cells showed that CD4^+^ T cells of VPM1002- and NUOG-vaccinated goats produced more IFN-γ compared to BCG-vaccinated and mock-treated animals. In summary, the subcutaneous application of VPM1002 and NUOG induced anti-tuberculous immunity, while exhibiting a comparable safety profile to BCG in goats.

## 1. Introduction

Tuberculosis (TB), caused by *Mycobacterium tuberculosis* (Mtb), is one of the leading causes of death in humans from a single infectious agent, currently surpassed only by the coronavirus (COVID-19) pandemic. In 2021, 10–11 million people worldwide fell ill with TB, and an estimated 1.6 million people died [1]. Other members of the Mtb complex (MTC), namely *M. bovis* and *M. caprae*, cause tuberculosis in domestic and wild ruminants. This is referred to as bovine TB (bTB), although *M. bovis* in particular shows a considerable host promiscuity [2,3]. About 50 million cattle globally are thought to be bTB positive, resulting in a calculated loss of 3 billion dollars annually [4]. In addition to posing a major economic and animal health problem, the causative agents of bTB are important zoonotic pathogens [5]. The World Health Organization (WHO) estimated that there were 140,000 new cases and 11,400 deaths of zoonotic TB due to *M. bovis* globally in 2019 [6]; it is even suspected that these numbers are significant underestimations. Consequently, the tripartite road map for zoonotic tuberculosis demands the availability and roll-out in endemic settings of effective anti-TB vaccines for people and livestock by 2025 [7].

*Bacille Calmette-Guérin* (BCG) is the only vaccine currently available against human TB, and it has been one of the most widely administered vaccines worldwide. In 2020, global BCG immunization coverage among 1-year-olds was estimated at 85% [8]. The efficacy of BCG is highly variable: it protects infants from severe forms of disseminated TB, but does not defend against the most common form, pulmonary TB [9]. Despite the increasing emergence of wildlife reservoirs, as well as the increasing interregional trade of animals and animal products, and the higher costs for disease control programs, no TB vaccine is yet approved for domestic livestock [10]. BCG was shown to significantly prevent formation of pathomorphological lesions in TB-infected cattle and goats, but, as in humans, the observed degree of protection varied [4]. Similarly important, the application of BCG interferes with common diagnostic methods applied in veterinary species and prescribed in TB control programs, such as the tuberculin skin test [10,11,12]. However, novel diagnostic approaches for veterinary application, corresponding to the Differentiating Infected from Vaccinated Animals (DIVA) strategy, are under development or have been experimentally applied [13,14,15]. In 2021, large scale field trials for the evaluation of the safety of a BCG vaccine and a DIVA-compliant skin test in cattle started in the United Kingdom (UK). If the trials are successfully completed, marketing authorization in the UK is envisaged by 2025 [16].

Several vaccine candidates, including recombinant live vaccines for BCG replacement and subunit vaccines (viral vectored or based on adjuvanted recombinant proteins) for boosting BCG responses are currently under development for human use [17]. One of the most advanced vaccine candidates in clinical trials is the recombinant live vaccine BCG Δ*ureC::hly* (VPM1002) strain. VPM1002 has been genetically modified to express *hly*, the gene encoding Listeriolysin O (LLO) of *Listeria monocytogenes*, in place of *ureC*, encoding the mycobacterial urease C. LLO is a major virulence factor that allows the escape of antigens from the phagosome into the cytosol by perturbating the phagosomal membrane [18]. LLO has an acidic pH optimum (<5.8). Upon infection of macrophages with tuberculous mycobacteria, urease C neutralizes phagosome acidification and inhibits phagolysosome maturation [19]. This prevents phagosome maturation and inhibits trafficking of MHCII to the cell surface, resulting in reduced antigen presentation. The deletion of *ureC* in VPM1002 thus has a double benefit. First, it allows acidification of the phagolysosome so that LLO can exert its effect. The resulting perturbation of the phagolysosomal membrane by LLO promotes egression of VPM1002-antigens into the cytosol, enhances cytosolic antigen loading via the MHC I pathway, and promotes cross-priming by induction of apoptosis of the infected macrophage [20,21]. Second, deletion of *ureC* allows phagolysosome maturation and trafficking of MHCII to the cell surface. Thus, the molecular alterations to VPM1002 aim to enhance both CD4^+^ and CD8^+^ T cell responses compared to the parental BCG strain. The clinical safety and immunogenicity of VPM1002 has been proven in humans [22,23,24], allowing for three ongoing phase III clinical trials (NCT 04351685, CTRI/2019/01/017026, NCT 03152903) [25]. To further improve the VPM1002 vaccine strain, two derivatives of the strain have been generated. The first derivative, VPM1002 ∆*pdx1* (PDX), is deficient in pyridoxine synthase, an enzyme that is required for the biosynthesis of the essential cofactor vitamin B6. In mice, adaptive immunity and protection induced by PDX have been shown to depend on increased dietary supplementation of vitamin B6. The possibility to control vaccine persistence through vitamin B6 levels aims to improve the safety profile of this vaccine candidate [26]. The second derivative, VPM1002 ∆*nuoG* (NUOG), was designed to further improve antigen presentation in comparison to VPM1002. Deletion of the anti-apoptotic virulence gene *nuoG*, which encodes NADH dehydrogenase I subunit G, increases host cell apoptosis and leads to increased targeting of the phagocytosed vaccine by the autophagosome marker LC3 [27]. In mice, this led to increased T cell responses and greater protection against *M. tuberculosis* challenge. The analysis of draining lymph nodes after vaccination demonstrated an earlier, stronger induction of immune responses by NUOG when compared to VPM1002 and parental BCG and suggested the upregulation of inflammasome activation and interferon-induced GTPases.

Goats are a potential target species, as well as suitable animal models for testing TB vaccines. As natural hosts, highly susceptible to *M. bovis* and *M. caprae*, they not only resemble cattle in terms of immune response and TB pathology, but they can also provide an important link between preclinical studies in murine species and subsequent testing in humans [28,29]. The TB disease pattern and pathology in goats are similar to those in humans, including slow disease progression, typical confinement to the respiratory tract, and formation of caseous granulomas. They better reflect the genetic background, environmental factors, and characteristics of the immune response of humans than do mice, but are more feasible in terms of ethical concerns and costs than non-human primates [5,29,30]. Because VPM1002 and its derivatives have not been assessed as to their safety and immunogenicity in ruminants or livestock until now, we aimed at deploying the caprine animal model for this purpose. Safety was assessed by clinical and hematological examination, observation of local reactions, and testing for vaccine shedding via the nose and for tissue distribution at necropsy. The immune response to the novel vaccine candidates was broadly monitored by means of antigen specific interferon gamma (IFN-γ) responses detected by interferon gamma release assay (IGRA) and by analysis of different T cell subsets and their antigen-specific IFN-γ production for 112 days after subcutaneous vaccination. Antibody responses against *M. bovis* antigens and cross-reaction to *M. avium* subsp. *paratuberculosis* (MAP) were analyzed, as well as the compatibility of the novel vaccine candidates with the DIVA strategy.

## 2. Results

### 2.1. Clinical, Hematological, and Bacteriological Findings Intra Vitam

The general health of the goats, assessed by daily clinical examinations, was not impaired by vaccination throughout the study. However, within the physiological range, the rectal temperature of VPM1002-vaccinated goats increased significantly at 6 days post vaccination (dpv) (median 39.2 °C) and 106 dpv (median 38.9 °C) and of NUOG-treated animals at 5 dpv (median 39.2 °C) and 106 dpv (median 39.7 °C). Elevated values at 106 dpv were recorded two days after purified protein derivative (PPD) application for the single intradermal comparative cervical test (SICCT; Figure 1).

Vaccination had no adverse effect on body weight gain. Differences between groups, apparent two weeks before vaccination, but not statistically significant, remained until the end of the experiment (Figure 2).

No significant differences between groups were noted for the differential white blood cell counts, except for the percentage of eosinophils at 0 dpv (median values: mock-treated group 1.0%, BCG 0.5%, VPM1002 3.0%, NUOG 3.0%, PDX 1.5%) and 84 dpv (median values: mock-treated group 1.5%, BCG 1.0%, VPM1002 2.5%, NUOG 3.5%, PDX 5.5%). The absolute number of eosinophils at 84 dpv (median values: mock vaccinated group 0.19 G/L, BCG 0.11 G/L, VPM1002 0.31 G/L, NUOG 0.53 G/L, PDX 0.70 G/L). Vitamin B6 serum levels were determined as 42.6 ± 11.1 µg/L (mean ± standard deviation (SD), *n* = 30) on 128 dpv, and did not differ between vaccination groups. Goats showed occasional to regular spontaneous coughing and modest nasal discharge prior to and throughout the study. These respiratory symptoms did not differ between groups, including the mock-treated control group, and slightly decreased over time. Goats did not shed the vaccine strains or any other mycobacteria by nasal secretion (Appendix A). Adverse effects of the vaccine candidates were restricted to local reactions. All animals vaccinated with BCG, VPM1002, NUOG, or PDX developed swellings and nodules at the vaccination site. The highest total lesion scores (mean total score per group considering all time points, as described in Materials and Methods) were exhibited by animals which had received NUOG (mean total score 4.2) and VPM1002 (mean total score 3.9), while the two other groups showed milder lesions (PDX 2.7, BCG 2.2; Table 1, Figure 3)

The nodules were solid, not movable in the subcutis, occurred particularly in the early stages of development, and were characterized by edematous swelling and pain. Approximately one month after vaccination, some goats of each group developed ulcerations in the overlying skin of the nodules (BCG 1/6, VPM1002 3/6, NUOG 2/6, PDX 1/6). The ulcerations of the two NUOG vaccinated goats were 5–10 mm in size (diameter) and were suppurating, finally forming a cavern. In the BCG-vaccinated goat, the ulceration was relatively smaller (3–5 mm diameter), also suppurating, but not cavernous. In the VMP1002- and PDX-vaccinated goats, the ulcerations were very small (2–3 mm diameter) and without exudation. In general, ulcerations were in the center of the nodules, and the area affected was very small in comparison to the sizes of the nodules, which were 2–5 cm in diameter for the PDX-vaccinated goat and 5–10 cm for the other animals at this time point. Exudation lasted for two or three days. Afterwards, all wounds were covered with a scab, which remained for up to three months. Viable vaccine strains were recovered from the exudate of one animal of the PDX and two animals of the NUOG group.

### 2.2. Post Mortem Findings

At necropsy (127 dpv), lesions were detected at the vaccination sites of all immunized, but none of the mock-treated, goats. Lesions consisted of well-demarcated nodules in the subcutis, where the vaccines had been injected, with the exception of one goat that had received PDX, which exhibited a connective tissue scar. Corresponding to the clinical findings, VPM1002-vaccinated goats had predominantly large nodules filled with white to yellow, creamy material, while BCG- and PDX-vaccinated goats had smaller nodules with firm central necrosis. Lesions in NUOG-vaccinated goats were of variable sizes and filled with white to yellow, creamy material. Histologic examination identified most of the nodules as type 3 granulomas, with central necrosis surrounded by a granulomatous inflammatory infiltrate. Acid-fast bacteria (AFB) were detected in all granulomas from BCG- and VPM1002-vaccinated goats and in four of the six goats vaccinated with PDX and NUOG, respectively. Details about morphological characteristics and cellular composition of these granulomas were recently reported [31]. Viable bacteria were recovered from the vaccination sites of three animals of the BCG group, two animals of the VPM1002 group, and one animal of the NUOG group (Table 2).

*Lymphonodi* (*Lnn.) cervicales superficiales* and *Lnn. axillares profundi*, which drain the vaccination site, were examined for lesions and their size compared with the lymph nodes from the contralateral site and from mock-treated goats. Superficial cervical lymph nodes had not changed in size, while the deep axillary lymph nodes were enlarged, especially in the goats which had been vaccinated with VPM1002. Granulomas were detected in some of the draining lymph nodes (dLNs) in BCG-, VPM1002- and PDX-vaccinated goats, but in none of the mock-treated and NUOG-vaccinated goats (Table 2). *Lnn. cervicales superficiales* were more activated than *Lnn. axillares profundi*. No increased activation was observed in lymph nodes with granulomas. Granulomas in BCG- and VPM1002-vaccinated goats were identified as type 3 granulomas, while those in PDX-vaccinated goats were type 1 granulomas characterized by an infiltrate of epithelioid cells and multinucleated giant cells. AFB were detected in most granulomas (both type 1 and 3), but cultural isolation of the vaccine strains was only successful in the mediastinal lymph node (without lesions) of one BCG-vaccinated goat (Table 2). The notion that the vaccine strains were predominantly contained locally was corroborated by the fact that macroscopic examination and bacteriological culture did not reveal granulomas and/or mycobacterial growth in any other tissues.

Background lesions detected in both mock-treated and vaccinated goats were mild to moderate periportal lympho-histiocytic infiltrates in the liver, mild multifocal interstitial lympho-histiocytic infiltrates in the kidney, and mild purulent tonsillitis and crypt abscesses. Chronic bronchitis and/or chronic bronchopneumonia and/or chronic fibrous pleuritis were seen in all mock-treated, 5 out of 6 BCG- or VPM1002-vaccinated, 3 out of 6 PDX-vaccinated and 2 out of 6 NUOG-vaccinated goats.

The presence of the vaccine strains or other mycobacteria in tissue samples was assessed by bacterial culture. *M. septicum*/*M. peregrinum* was isolated from the contralateral skin site of one mock-treated goat. *M. gordonae* was detected in the mesenterial lymph nodes and/or Peyer´s patches of four different goats from different groups (Appendix A). Neither vaccine strains nor MAP were detected in any of the tissue samples.

### 2.3. Cell-Mediated Immune Response Assessed by SICCT and IGRA

A delayed type hypersensitivity reaction, analysed by SICCT on 104 dpv, was induced in all goats vaccinated with BCG, VPM1002, NUOG, and PDX, but not in the mock-treated group. The extent of the response did not differ markedly between the four vaccinated groups (Figure 4). The increase in skin thickness after application of bovine PPD (bPPD) in vaccinated animals was several times higher than after application of avian PPD (aPPD), and the difference exceeded the threshold value for a positive test result (ΔbPPD–ΔaPPD > 4 mm) in the in vivo diagnosis of bovine tuberculosis.

Immunization with the vaccine candidates and BCG elicited a distinct antigen-specific IFN-γ response after ex vivo stimulation of whole blood samples. All vaccine candidates induced a responsiveness of the vaccinees to both antigens (Figure 5a,b), with bPPD-induced IFN-γ levels exceeding the aPPD-induced levels at each time point examined. The responses of the BCG-, VPM1002- and NUOG-vaccinated groups peaked directly after vaccination (28 dpv) and after SICCT testing (112 dpv) and decreased between 56 and 84 dpv, particularly after re-stimulation with aPPD. The response of the PDX group was lower than that of the other three groups at 28 and 56 dpv, and significantly differed from that of the mock-treated group from 56 dpv onwards in the bPPD-stimulated approach. When the difference in the response to bPPD and aPPD was assessed, a positive response (ΔΔIFN-γ > 0.1) was already observed at 28 dpv in almost all goats vaccinated with BCG, VPM1002, or NUOG. Statistically significant differences compared with the mock-treated group became evident on 56 dpv until the end of the experiment (Figure 5c). The ΔΔIFN-γ response of PDX-vaccinated goats significantly differed from mock-treated goats on 84 dpv, and Δ corrected optical density (cOD) values only exceeded 0.1 in some animals. Stimulation with the *M. bovis* specific peptide cocktails Bovigam^®^ PC-EC and PC-HP (Prionics Lelystad B.V., Lelystad, The Netherlands) did not result in a significant IFN-γ response in whole blood from goats of any vaccination group, analysed on 28 dpv and 84 dpv (Figure 6).

### 2.4. Cellular Immune Responses Assessed by Quantitation of Intracellular IFN-γ in T Cell Subsets after In Vitro Stimulation

Vaccination with BCG, VPM1002, NUOG, and PDX did not affect the overall proportions of CD4^+^, CD8^+^, and γδ T cells in peripheral blood mononuclear cells (PBMC) isolated from the animals (Appendix A). In vitro stimulation of PBMC with bPPD induced an increase in the proportion of IFN-γ-producing lymphocytes, compared to PBMC cultured in medium only, in all groups (Figure 7). Analysis of the relative proportion of lymphocytes producing intracellular IFN-γ among total lymphocytes showed a distinct increase in vaccinated animals compared to mock-treated animals on 56 dpv, which was statistically significant for the BCG- and VPM1002-vaccinated groups only. Analyzing the proportions of IFN-γ^+^ lymphocytes within the CD4^+^, CD8^+^, and γδ T cell subsets revealed an increase in NUOG- and/or VPM1002-vaccinated goats compared to BCG and the mock-treated group at several time points, although the differences were not statistically significant (Appendix A).

Group specific differences became evident when the relative amount of intracellular IFN-γ, determined as median fluorescence intensity (MFI), of stimulated immune cell subsets upon culture in the presence of antigen was considered (Figure 8). The analysis revealed significant differences for the CD4^+^ populations at 56 dpv and for γδ T cells at 84 dpv. At 56 dpv, the CD4^+^ T cells of VPM1002- and NUOG-vaccinated goats showed increased IFN-γ production, compared to BCG and mock-treated animals. The PDX-vaccinated group showed no differences in comparison to the mock-treated or the other three groups. At 84 dpv, the γδ T cells of all vaccinated groups had higher MFI values than did the mock-treated group. CD8^+^ cells also showed a tendency to increased responsiveness in the vaccinated animals from 84 dpv onwards, but the effect was veiled by profound biological variation within the animal groups.

### 2.5. Humoral Immune Responses

Antibodies against *M. bovis* antigens MPB70 and MPB83 were not detectable in any of the animals at any time point during the experiment (Figure 9a). In contrast, moderate to high amounts of cross-reacting antibodies against MAP antigens were measurable at 28 dpv and thereafter until the end of the experiment (Figure 9b). Vaccination with all three vaccine candidates and BCG significantly increased the response to MAP antigens in comparison to that of the control group from 28 dpv to 84 dpv. On 28 and 56 dpv, the NUOG- and the VPM1002- vaccinated groups showed significantly higher cOD values than did the BCG and the PDX groups. The highest antibody levels were observed in all vaccinated groups on 112 dpv, 7–8 days after the intradermal application of PPDs for the SICCT. At this time point, the level of cross-reacting antibodies was the highest in the NUOG- and BCG-vaccinated animals and the lowest in the PDX-vaccinated goats. VPM1002 did not differ from the control group on 112 dpv, due to a high individual variation. 

## 3. Discussion

In response to the demand for improved TB vaccines, a century-old vaccine strain, BCG, was modified to generate strain VPM1002, which was found to be safer and more effective than BCG in preclinical studies [18,32,33,34,35], as well as safe and immunogenic in humans [22,23,24]. The two derivatives NUOG and PDX, developed to further increase efficacy or safety, respectively, have only been tested in murine models up until now. This study aimed at assessing these promising human TB vaccine candidates for use in ruminants. Data regarding the safety and immunogenicity of the novel vaccine candidates, compared to BCG, will contribute to TB vaccine development for both human and veterinary applications and provide the basis for forthcoming efficacy studies.

The monitoring of clinical signs after subcutaneous vaccination of the goats did not reveal any systemic reactions. A prominent adverse event induced by VPM1002, NUOG, PDX, and BCG was the development of lesions at the site of injection. The largest lesions were caused by VPM1002 and NUOG, whereas those in PDX- and BCG-vaccinated goats were smaller. At necropsy (16 weeks post vaccination (pv)), lesions were confirmed to be circumscribed vaccine granulomas, with central necrosis in the subcutis, and the re-isolation of viable vaccine strains was successful in some goats of all vaccinated groups, except for the PDX group. A detailed histological characterization of these vaccination granulomas has recently been published [31]. Vaccine granulomas have been reported as sequelae of subcutaneous vaccination with BCG, both in adult goats and in goat kids [36]. In the present study, granulomas examined at 24 weeks pv were smaller than those at 16 weeks pv, indicating that local lesions slowly regressed with time. Vaccine granulomas in goats occur not only after vaccination with BCG, but also after other vaccinations, e.g., against paratuberculosis [37,38]. Local and regional reactions, such as abscesses and lymphadenitis, are the most common adverse events of BCG in many animal species and humans [10]. BCG vaccine-induced local granulomas have been recognized in humans for many years [39], and are characteristic of immunocompetent individuals [40]. Erythema and induration at the site of injection were the most commonly observed adverse events in humans after vaccination with BCG or VPM1002 [22]. The number of these, primarily local, adverse events after VPM1002-vaccination was similar to or lower than those noted after BCG-vaccination in BCG-naive people [22]. At necropsy, granulomas were also found in the dLNs of a few of the VPM1002-, PDX- and BCG-vaccinated goats, but in none of the NUOG-vaccinated goats. No dLNs contained viable bacteria of the respective vaccine strains. Retrieval of mycobacterial vaccine strains may be time-dependent, since isolation of BCG has been described from the dLN of a goat kid euthanized at 8 weeks pv, but not from goats at 24 weeks pv [36]. Apparently, viable bacteria are eliminated by local immune responses faster than the granulomas regress, resulting in sterile granulomas.

In mice subcutaneously vaccinated with BCG, VPM1002, NUOG, or PDX, all vaccines disseminated to some extent to the dLNs and to the spleen. Dissemination of VPM1002 and NUOG was significantly reduced in comparison to BCG, and clearance from the dLNs was faster [27,33]. Persistence of PDX in the dLNs and spleen in mice not supplemented with Vitamin B_6_ was extensively reduced in comparison to BCG and VPM1002. Vitamin B_6_ supplementation restored the extent of dissemination to the same level as that of VPM1002 in dLNs, but not in the spleen [26]. In accordance, the data from goats suggest that VPM1002 and its derivatives retained the potential of BCG to disseminate to the dLNs, but granuloma development at these sites may have contributed to limiting of further spreading. As in mice, we found no evidence of VPM1002, NUOG, and PDX disseminating to any other organs [26,27,33,41,42]. Only BCG could be re-isolated from the mediastinal lymph node of one goat. Goats did not shed the vaccine strains, e.g., via nasal secretion, in this study, nor by milk and feces after vaccination with BCG [36]. Similarly, humans were not found to shed VPM1002 through nasal discharge, saliva, urine, or stool after vaccination [22]. However, subcutaneous vaccine granulomas showed superficial ulceration and suppuration in some animals. Viable mycobacteria were re-isolated from granuloma discharge of 2/6 NUOG-vaccinated and 1/6 PDX-vaccinated goats, which could have led to contamination of the environment. VPM1002 and its derivatives are genetically modified organisms which require special considerations for licensing as vaccines for use in livestock destined for human consumption. However, VPM1002 has a superior safety profile [18,20,27], which has allowed for its use in several human clinical trials as a TB vaccine [20,22,23] and for supportive treatment of bladder cancer [20,43]. In the latter setting, even shedding of the vaccine strains by emiction is considered sufficiently safe for patients, household members, and the environment.

The immunogenicity of the novel vaccine candidates in goats was proven by means of the IGRA and SICCT. VPM1002, NUOG, and BCG elicited strong IFN-γ responses and at comparable levels, although with high individual variation, already noticeable at 28 dpv and lasting until the end of the experiment. The antigen-specific IFN-γ production of ruminants, notably cattle and goats, in response to different BCG strains [44,45], different deletion mutants of *M. bovis* [46,47], or different vaccination regimens, including prime-boost strategies with BCG and virally vectored booster vaccines [48], has been reported. Generally, marked IFN-γ responses were induced, measurable as early as 2–3 weeks post vaccination. The time course and magnitude of the IFN-γ response varied between the studies, and it became obvious after challenge with virulent *M. bovis* that the antigen-specific IFN-γ response was not suitable as a surrogate for or correlate of protection [45]. Grode et al. [22] showed that vaccine dose impacts on the magnitude and the time course of the immune response induced. In the present study, actual subcutaneously applied doses varied between 1.6 ± 1.1 × 10^7^ colony forming units (CFU) per goat (mean ± SD, BCG-vaccinated group) and 13.4 ± 5.8 × 10^7^ CFU per goat (NUOG-vaccinated group). Buddle et al. [49] found no differences between groups of cattle vaccinated with either 1–4 × 10^5^ CFU or 1–4 × 10^6^ CFU BCG Danish regarding the kinetics and magnitude of the IFN-γ and SICCT response. In deer, the extent of the immune response quantified by a lymphocyte transformation assay and a tuberculin skin test correlated to the dose of BCG Pasteur. In comparison to a low (5 × 10^4^ CFU) and medium (5 × 10^5^ CFU) dose group, the high dose group (5 × 10^8^ CFU) showed a bias towards type 2 rather than type 1 immune responses [50]. As intra-individual variation exceeded the differences between groups in the present study, and the extent of the IFN-γ and SICCT response did not differ between the four vaccines, except for PDX, it appears that the variation in dosages did not impact the comparability of the results. Compared to the other three vaccine strains, PDX showed a delayed TB-specific IFN-γ response, reaching IFN-γ levels only similar to those of the BCG or VPM1002 group at 112 dpv. PDX was designed to show an improved safety profile based on the assumption that it will succumb at limiting vitamin B_6_ levels. The present data reaffirmed that PDX is also attenuated in ruminants. The metabolism of ruminants differs from that of other mammals, particularly in that essential nutrients, such as vitamins, are acquired from bacterial sources in their gastrointestinal tract rather than from the diet itself. The serum vitamin B_6_ levels of the goats, which were not dietary supplemented, were in the range of 200–300 nmol/L, slightly above the reference limits for human adults, i.e., 23–223 nmol/L [51]. Established reference intervals for mice are missing, but Hsu et al. [52] determined a mean value of 243.98 ± 30.36 nmol/L, obtained from the plasma of nine mice, which is in the same range as the values we determined in goats.

The adaptive immune response of ruminants to TB, including the roles of T cell subsets (e.g., CD4^+^, CD8^+^, and γδ TCR^+^) and the function and sources of IFN-γ, is quite similar to that in humans [53]. In both humans and ruminants, an essential component of the immune response to BCG vaccination and TB infection is the production of IFN-γ by T helper 1 CD4^+^ T cells [54]. IFN-γ producing CD8^+^, γδ TCR^+^, and natural killer (NK) cells also play an important role [53]. In contrast to humans and mice, cattle and other ruminants have very high proportions of γδ T cells. In cattle, γδ T cells play a crucial role in the immune response against bovine TB [55]. However, the role of γδ T cells in caprine TB remains to be elucidated [56]. Goats naturally infected with *M. bovis* did not show elevated proportions of γδ T cells in peripheral blood, lungs, mediastinal lymph nodes, or other lymphoid organs [57,58]. VPM1002 induced a quantitative increase in mycobacteria-specific CD4^+^ T cells in comparison to BCG in mouse lymph nodes and spleens [33]. However, a phase I study found no major differences in the antigen-induced cytokine secretion by circulating CD4^+^ and CD8^+^ T cells between VPM1002- and BCG-vaccinated humans [22]. In goats, the percentages of circulating antigen-specific IFN-γ+ T cells were increased in the vaccinated groups at 56 dpv, reaching significance only in the BCG- and VPM1002-vaccinated groups. Furthermore, the VPM1002- and NUOG-vaccinated groups showed a tendency to higher MFI values for IFN-γ in CD4^+^ T cells at 28, 56, and 84 dpv, although this was only statistically significant at 56 dpv. In contrast, IFN-γ MFI values tended to be elevated later in CD8^+^ T cells, at days 84 and 112, in all vaccinated groups. At 84 dpv, IFN-γ MFI values in γδ T cells were higher in all vaccinated groups compared to those in the mock-treated group. As the live vaccines primarily induced a local inflammatory response, with only some dissemination to the dLNs, the analysis of the low numbers of circulating antigen-specific cells may be insufficient for detecting differences between individual vaccine strains. In addition, minor effects could be masked by the limited group size feasible in large animal trials and the inherent biological variation when outbred animals are tested.

The responses of the goats in SICCT, a variant of the tuberculin skin test, and IGRA, after vaccination with the novel vaccine candidates, imply that these vaccines, such as BCG, could significantly interfere with routine diagnostics applied in veterinary medicine. The tuberculin skin test, sometimes accompanied by IGRA, is still the primary screening method for bTB [59], both deploying PPD antigens by default. While all vaccinated goats significantly reacted in those tests, none of the animals responded to peptide cocktails, based on ESAT-6 and CFP-10 antigens, in the IGRA tests. As diagnostic methods for application in veterinary species, based on these antigens, are currently under development [15,16], introduction of the novel vaccine candidates for use in ruminants is considered compatible with the DIVA concept. The *M. bovis* antigens MPB70 and MPB83 might also serve as DIVA antigens because they are produced only meagerly by some BCG strains, including BCG Danish strain [60]. In cattle, specific antibodies after BCG vaccination are rarely observed (reviewed by [61]). Consistently, no antibodies against *M. bovis* antigens were detected in any of the goats. In mice, VPM1002 and NUOG induced higher antibody levels compared to BCG [27].

The application of the vaccine candidates in goats interfered with MAP serology. All animals are considered negative for MAP, as the agent was not cultured from tissues of any of the goats at necropsy, and the mock-treated animals did not develop antibodies against MAP during the course of the experiment. We concluded that the vaccine candidates induce seroconversion to MAP antigens, albeit the attenuated PDX was less effective in this regard. These cross-reactive antibody responses to MAP seemed to be boosted by the intradermal application of PPDs for the SICCT at 104 dpv in all vaccinated groups. Thus, these unintended immunological reactions have to be carefully considered.

Some limitations of the study, which reflect the peculiarities of large animal models in general, are the increased variability, due to the small group size feasible in experimental settings, and the genetic diversity of outbred species, as well as the almost unavoidable pre-exposure to a variety of pathogens. As goats bred specifically for research purposes are not available in Germany, the goats were purchased from a conventional farm. The animals showed mild respiratory symptoms, which were treated and improved prior to the experiments, but induced minor pathomorphological lesions detectable at necropsy. Respiratory infections, often sub-clinical or caused by opportunistic bacteria, are frequently observed in goat production. Pathomorphological background lesions could be clearly distinguished from lesions typically induced by BCG or its derivatives and, like the clinical symptoms, were seen in a few animals in all groups. Conclusions of the study were based on the comparison between goats that had received the different vaccines or were mock-treated. Although we cannot rule out an influence on any of the immune parameters analyzed, pre-exposure to pathogens reflects the situation in the desired target population.

In summary, the results of the study suggest that VPM1002 and its derivatives are immunogenic and despite local adverse effects, are sufficiently safe and DIVA-compatible to be further explored for veterinary applications. Nevertheless, as there are no reliable immune correlates of protection (CoP), the efficacy of the novel candidates in preventing or ameliorating bovine TB infections in goats must be evaluated in challenge trials, which are currently underway.

## 4. Materials and Methods

### 4.1. Experimental Animals, Housing, and Health Status

Thirty male goats of the “German Improved White” breed were purchased from a conventionally raised herd of dairy goats with no history of bovine TB. The goats had been castrated and vaccinated against *Clostridium* spp., *Mannheimia haemolytica*, and *Pasteurella trehalosi* (Heptavac^®^ P plus, Intervet Deutschland GmbH, Unterschleißheim, Germany) in their herd of origin. After arrival at the experimental animal facility, they were accommodated for three weeks before first sampling and five weeks before vaccination with the novel vaccine candidates. The animals were kept in groups, in air-conditioned loose-boxes with natural daylight, bedded on straw, and fed with hay and age-adjusted portions of concentrated feed (Alleinfuttermittel für Ziegenlämmer, LHG-Landhandelsgesellschaft eG, Schmölln, Germany). Water was provided ad libitum. During this acclimatization phase, the actual health status of the goats was determined. Most of the animals showed intermediate or regular spontaneous coughing and low-grade nasal or conjunctival discharge. Routine screening was performed for bacterial infections, including infections with *Salmonella* spp., *Mycoplasma* spp., *Pasteurella* spp., *Coxiella burnetti*, *Chlamydia* spp., as well as for ecto- and endoparasites. Most of the experimental animals shed *Eimeria* spp. (23 goats out of 30). Bacteriological examination of the nasal swabs showed the presence of *Mannheimia haemolytica* (6/30), *Moraxella bovis* (4/30), and *Staphylococcus aureus* (3/30). Almost all goats were polymerase chain reaction (PCR)-positive for *Mycoplasma* spp. (27/30), but negative in culture. The examination for *Salmonella* spp. (bacterial culture of fecal swabs), *Pasteurella* spp. (bacterial culture of nasal swabs), *Coxiella burnetti* (serum antibodies), and *Chlamydia* spp. (serum antibodies) yielded negative results throughout. Four weeks before experimental vaccination, all goats received antibiotic treatment for 5 days with Enrofloxacin (Baytril^®^, Bayer, Leverkusen, Germany) and a single treatment against coccidiosis with Toltrazuril (Baycox^®^, Bayer). Clinical symptoms improved before the beginning of the experiments; nevertheless, goats of all groups, including the controls, showed mild signs of chronic bronchopneumonia, chronic bronchitis, or pleuritis at necropsy. The herd of origin had tested positive for MAP in the past. However, the analysis for MAP of necropsy tissue samples of the goats, by mycobacterial cultivation and subsequent MAP-specific PCR of visible colonies, remained negative. Thus, the animals included herein are considered free of active paratuberculosis.

The data presented by our group in a prior publication [31] and the current manuscript are based on samples obtained from the same cohort of animals. We consider this an important implementation of the 3R principles (replacement, reduction, and refinement) for the use of experimental animals. However, there is a clear distinction between the experimental approaches used in these papers: While in this manuscript results pertaining to systemic immunity and safety are reported, our previous paper, by Liebler-Tenorio et al., focuses on the detailed morphologic analysis of the vaccine granulomas.

### 4.2. Vaccination, Clinical Examination, and Intra Vitam Sampling

At an average age of five months (mean ± SD: 150 ± 12 days, minimum: 134 days, maximum: 184 days), four groups of goats (*n* = 6) were vaccinated with either one of the vaccine candidates, BCG Δ*ureC::hly* (designated as VPM1002), PDX, NUOG, or with BCG SSI. A fifth, mock-treated group (*n* = 6) served as a control group. The goats´ age and body weight at the time of vaccination did not significantly differ between the five groups. To avoid cross-contamination, each group was housed in a separate room, and a strict hygienic regime was deployed, including disinfection of materials and change of clothes when moving from one group to another. Cryopreserved aliquots of the tuberculosis vaccine candidate VPM1002 and its derivatives, PDX and NUOG, as well as BCG strain SSI, were provided by the Max Planck Institute for Infection Biology (Berlin, Germany). The original stock of BCG SSI 1331 Danish ATCC 357533 was obtained from American Type Culture Collection (Manassas, VA, USA). The mycobacteria stocks were washed three times with phosphate-buffered saline solution (PBS, Thermo Fisher Scientific, Waltham, MA, USA) and reconstituted in PBS, with an intended bacterial count of 5 × 10^5^ CFU per dose. The vaccine candidates were administered subcutaneously behind the left scapula. Prior to vaccination, the area was shaved, checked for skin lesions, and disinfected with an alcohol swab. Each animal received a total volume of 500 µL, including the control group receiving the same amount of PBS. The vaccines were applied at actual total doses (determined by re-titration) of 7.3 ± 4.7 × 10^7^, 3.7 ± 1.5 × 10^7^, 13.4 ± 5.8 × 10^7^, and 1.6 ± 1.1 × 10^7^ CFU (mean ± SD of three independent cultures) per goat for VPM1002, PDX, NUOG, and BCG, respectively.

A comprehensive clinical examination of the animals was carried out daily, comprising general condition, appetite, rumination, skin and hair coat, eye and nasal discharge, respiratory and heart rate, coughing (assessment of quality; spontaneous and evoked), lung and abdominal auscultation, rectal temperature, consistency of feces, examination of extremities, and size and painfulness of lymph nodes. All examinations were documented using a scoring system. Body weight was taken in weekly intervals. Post vaccination, injection sites were observed daily for signs of local inflammatory reactions, including the assessment of the lesion size, redness, pain, swelling, local temperature, and necrosis (Table 3). The scoring system is based on the expectable adverse effects, induced by BCG and other mycobacterial live vaccines, in accordance with their effect on the goats´ general health and wellbeing.

Blood sampling was performed 14 days and immediately before vaccination (day 0) and every 4 weeks until 112 dpv. To reduce variation, regular sampling and vaccination, was performed in a staggered mode. One or two goats of each group were sampled per week, resulting in each goat being sampled once a month, but all animals were sampled at the same dpv. Ethylenediaminetetraacetic acid (EDTA) blood, heparinized blood, and serum were obtained from the external jugular vein. Additionally, nasal swabs were taken for mycobacterial culture every 4 weeks.

### 4.3. Blood Count and Vitamin B6 Levels

A differential white blood cell count, including the absolute and relative numbers of total leukocytes, neutrophils, lymphocytes, monocytes, eosinophils, and basophils, was performed by manual differentiation from EDTA blood (S-Monovette^®^, Sarstedt AG & Co. KG, Nümbrecht, Germany). For staining, Leuko-TIC^®^ (Bioanalytic GmbH, Umkirch, Germany) and HemaDiff quick stain (Bioanalytic GmbH) were used. Pyridoxal phosphate (vitamin B_6_) serum levels were quantified on 128 dpv by high performance liquid chromatography (HPLC) in an external diagnostic laboratory (Biocontrol, Mainz, Germany).

### 4.4. Interferon Gamma Release Assay (IGRA)—ELISA

For the IGRA, heparinised blood was stimulated overnight with 300 IU bPPD/mL (Prionics Lelystad B.V., Lelystad, Netherlands) or with 250 IU/mL aPPD (Prionics Lelystad B.V.). Pokeweed mitogen stimulated blood (5 µg/mL, Thermo Fisher Scientific) served as the positive control, and blood complemented with cell culture medium only (RPMI 1640 W/GLUTAMAX I, Sigma Aldrich Chemie GmbH, Taufkirchen, Germany) served as the negative control. Blood samples taken at 0, 28, and 84 dpv were additionally stimulated with antigens Bovigam^®^ PC-EC (Prionics Lelystad B.V.), which is based on ESAT-6 and CFP-10 antigens, and Bovigam^®^ PC-HP (Prionics Lelystad B.V.), with peptides derived from ESAT-6 and CFP-10, RV3615c, OmpA, and two additional antigens. IFN-γ in supernatants was quantified by in-house capture ELISA using monoclonal antibodies against bovine IFN-γ (capture antibody: clone CC330, Bio-Rad, Hercules, CA, USA; detection antibody: clone CC302-Biotin, Bio-Rad), horseradish peroxidase- (HRP) labeled streptavidin as a conjugate (Bio-Rad), and 3,3′,5,5′-tetramethylbenzidine (TMB) as the substrate (Merck, Darmstadt, Germany). Results were expressed as cOD values. Delta optical density (ΔOD) values were generated by subtraction of the cOD values for medium controls (ΔbPPD = cOD bPPD − cOD medium control; ΔaPPD = cOD aPPD − cOD medium control). The IFN-γ response was expressed as delta delta optical density (ΔΔIFN-γ), i.e., the difference between data obtained after stimulation with bPPD and aPPD (ΔOD bPPD − ΔOD aPPD) was calculated [62]. 

### 4.5. Intracellular IFN-γ of Different T Cell Subsets (Flow Cytometry)

The IFN-γ response of different T cell subsets was analysed by flow cytometry. PBMC were isolated by density gradient centrifugation (Pancoll Animal, density 1.077, Pan Biotech GmbH, Aidenbach, Germany) from heparinized blood samples (heparin sodium salt 75 IU/mL, Sigma Aldrich Chemie GmbH) and washed three times with Hanks Buffer without calcium and magnesium (in-house). Cells obtained from the interphase were resuspended at a final concentration of 2 × 10^6^ PBMC per ml in cell culture medium (RPMI 1640 W/GLUTAMAX I, Gibco^®^, Thermo Fisher Scientific), with 10% fetal calf serum (Gibco^®^, Thermo Fisher Scientific), 10 mM HEPES (Biochrom, Berlin, Germany), and 1% Penicillin/Streptomycin (Biochrom) were added. Ex vivo stimulation was conducted with 300 IU/mL bPPD (Prionics Lelystad B.V.). Unstimulated cultures incubated in medium and cultures supplemented with 1 µg Concanavalin A (Merck) were included as negative and positive controls, respectively. All cultures were incubated for 21 h at 37 ± 2 °C, 5% CO_2_ with Phorbol 12-Myristate 13-Acetate (PMA, 50 ng/mL, Merck), Ionomycin (1 µg/mL, Merck), and Brefeldin A (1 µL/mL, Golgi Plug^TM^, BD Biosciences, San Jose, CA, USA) added for the last 3 h of incubation.

Intracellular IFN-γ was quantified in CD4^+^, CD8^+^ and γδ T cells at the single cell level by flow cytometry. Samples were incubated with anti-CD4 (clone GC1A, Kingfisher Biotech, Saint Paul, MN, USA), followed by anti-mouse IgG_2a_ Alexa fluor 633 (Southern Biotech, Birmingham, AL, USA) secondary antibody, and Phycoerythrin (PE)-labelled anti-bovine IFN-γ (clone CC302, Kingfisher Biotech). Alternatively, cells were labeled with anti-CD8α (MCA2216GA, Biorad, Hercules, CA, USA) and anti-γδ TCR1-N24 δ chain (clone GB21A, Kingfisher Biotech), followed by anti-mouse IgG_2a_ Alexa fluor 633, anti-mouse IgG_2b_-FITC (Southern Biotech) and anti-IFN-γ-PE (as above). All primary antibodies were monoclonal, while secondary antibodies Alexa fluor 633 and FITC were polyclonal. After incubation with primary antibodies, the PBMC were fixed and permeabilized with BD Cytofix/Cytoperm (BD Biosciences), according to the instructions of the manufacturer, before staining with secondary antibodies and PE-labeled anti-IFN-γ was performed. Samples were analyzed with FACS CANTO II (BD Biosciences). Data analysis was performed with FlowJo^TM^ (version 10.6.2, BD Biosciences). After gating for live and morphologically intact lymphocytes (Figure 10a), results were expressed either as proportions of a specific population (e.g., percentages of the lymphocyte population, Figure 10b) or as median fluorescence intensity (MFI) values. Percentages of IFN-γ^+^ cells, referring to a T cell subpopulation (CD4^+^, CD8^+^ or γδ T cells), were obtained according to Figure 10c. The MFI for IFN-γ was derived from the whole, i.e., IFN-γ-positive and -negative, subpopulation, because this allows for the detection of even minor shifts in the activation-dependent expression of IFN (Figure 10d). As a measure of the antigen specific effect for all analytes, the p/u ratio was calculated by the quotient of the results for bPPD stimulated cells and the results for unstimulated cells:$\;\left[\frac{\%\;or\;MFI\;bPPD\;stimulated\;cells\;\left(p\right)}{\%\;or\;MFI\;unstimulated\;cells\;\left(u\right)}\right]$.

### 4.6. Antibody Response against M. bovis and M. avium subsp. Paratuberculosis (MAP)

Serum antibodies against *M. bovis* were detected with a modified IDEXX *M. bovis* antibody ELISA (IDEXX, Westbrook, ME, USA). The test is designed to detect antibodies against MPB70 and MPB83. The IgG HRP conjugate of the kit was replaced by anti-ruminant IgG HRP conjugate from the IDEXX Paratuberculosis Screening ELISA (IDEXX, Montpellier, France). The occurrence of cross-reactive antibodies with MAP was examined with the ID Screen Paratuberculosis Indirect Screening ELISA (IDvet, Grabels, France). Except of the mentioned modification, both test kits were applied following the manufacturer instructions.

### 4.7. Single Intradermal Comparative Cervical Test (SICCT)

A SICCT test was performed approximately one week prior to necropsy, at 104 dpv. On the right side of the neck, 0.1 mL (5000 IU) bPPD (WDT, Garbsen, Germany) was injected intracutaneously. On the left side of the neck, 0.1 mL (2500 IU) aPPD (WDT) was applied. Application and analysis were performed according to the tuberculosis regulation of the European Commission (annex B to council directive 64/432/EEC).

### 4.8. Necropsy

On 127 dpv (mean; minimum 119 dpv, maximum 135 dpv), animals were sacrificed by intravenous injection of 100 mg/kg Pentobarbital-sodium (Release 500 mg/mL^®^, WDT) after prior sedation by intramuscular injection of 0.25 mg/kg Xylazin (Rompun^®^ 2%, Provet AG, Lyssach, Switzerland). A complete necropsy with macroscopic assessment was performed. For histologic evaluation, samples were collected from vaccination sites, including skin from the left and the contralateral right site, *Lnn. cervicales superficiales* (left and right), *Lnn. axillares profundi* (left and right), tonsils, *Lnn. retropharyngeales*, thymus, lungs, *Lnn. mediastinales*, *Lnn. tracheobronchiales* (left), heart, liver, *Lnn. hepatici*, spleen, kidneys, *Lnn. renales*, adrenals, Peyer´s patches, *Lnn. mesenteriales*, *Lnn. ileocolici*, and sites with macroscopic lesions. Tissues were embedded in paraffin and paraffin sections stained with hematoxylin and eosin for histologic evaluation. In addition, sections from vaccination sites and dLNs were stained using the Ziehl–Neelsen method for the detection of AFB. Granulomas were classified according the classification scheme for tuberculous granulomas in ruminants by Wangoo et al. [63]. Samples from the injections sites of the animals were also used for a detailed morphological investigations of vaccine granulomas [31].

### 4.9. Bacterial Examination of Nasal Swabs and Tissue Samples

Nasal swabs and tissue samples taken at necropsy were examined by mycobacterial culture to assess the dissemination of the vaccine candidates within the body. Tissue samples included skin at the vaccination and contralateral site, *Lnn. cervicales superficiales*, *Lnn. axillares profundi*, lungs, *Ln. tracheobronchialis sinister*, *Lnn. mediastinales*, liver, spleen, *Lnn. hepatici*, kidney, *Lnn. renales*, heart, tonsil, *Lnn. retropharyngeales*, Peyer´s patches, *Lnn. mesenteriales craniales*, *Lnn. ileocolici*, bone marrow, and lung lesions, if present. From each tissue sample, 1 g was placed into 10 mL PBS and homogenized for 6 min at room temperature using a stomacher. A total of 10 mL of NALC-NaOH solution (containing 5 mL NaOH 4%, 5 mL sodium citrate dihydrate 2.9%, and 0.5 g N-acetyl-L-cysteine; all Sigma Aldrich Chemie GmbH) were added, and the sample was agitated for 25 min at 300 rpm on a shaker. After addition of 20 mL PBS, the sample was vortexed and centrifuged for 20 min at 3800× *g*. The supernatant was discarded, 10 mL PBS added and the sample centrifuged as before. The supernatant was discarded, the pellet resuspended in 1 mL PBS, and homogenized thoroughly by vortexing. A total of 200 µL of each resuspended pellet were transferred to 1 slant of Löwenstein–Jensen medium with Polymyxin B, Amphotericin B, Carbenicillin, Trimethoprim (PACT), and Glycerin and to 2 slants of Coletsos medium with PACT (both Artelt Enclit GmbH, Wyhra, Germany). In addition, one slant of Herrold’s Egg Yolk Medium with Mycobactin J (Becton Dickinson, Heidelberg, Germany) was inoculated with 200 µL of the pellets from the Peyer´s patches and *Lnn. mesenteriales craniales* and *Lnn. ileocolici*. The tubes were incubated for one week in a horizontal position, and afterwards, in an upright position for up to 12 weeks. Cultures were checked every 2 weeks for colony growth. When visible colonies appeared, the presence of *M. bovis* was confirmed by real-time PCR targeting IS*1081* [64] and by endpoint PCR targeting RD4 [65]. The PCR-based genotyping of VPM1002 and its derivatives was performed using the primers listed in Appendix A). When neither *M. bovis* nor VPM1002 and its derivatives could be confirmed in the colony material, mycobacterial species identification was performed by conventional PCRs targeting the IS*1245* and IS*901* for members of the *M. avium* complex [66,67] and IS*900* for *M. avium* subsp. *paratuberculosis* [68]. Mycobacterial species not identifiable by these PCRs were identified by nucleotide sequence analysis (GATC, Konstanz, Germany) of a PCR-generated DNA fragment of the 16S rRNA gene [69].

Nasal swabs were soaked in 10 mL PBS containing 1 mL PANTA (Polymyxin B, Amphotericin B, Nalidixic acid, Trimethoprim, Azlocillin; all Sigma Aldrich Chemie GmbH) for 48 h at room temperature. After thorough vortexing, the swab was removed, 10 mL of NALC-NaOH solution was added, and the sample was further processed, as described above. 

### 4.10. Statistical Analysis

Statistical analyses were calculated using R (version 4.0.2, R Foundation for Statistical Computing, Vienna, Austria). Group differences were determined with the non-parametric Kruskal–Wallis test, with a significance level determined at 0.05. In case the *p*-value indicated that there were group differences between at least two of the groups and the null hypothesis needed to be rejected (*p* ≤ 0.05), a pairwise comparison between the different vaccination groups was conducted. For the pairwise comparison, the Mann–Whitney-U test was used, with a significance level determined to be 0.05 (*p* ≤ 0.05). All data, except for body temperature and body weight, were plotted as boxplots showing the median (horizontal line), the interquartile range (box), and minimum and maximum (whiskers). All data beyond 1.5 times the interquartile range are depicted as outliers (dots). For clinical data, which was assessed daily, statistics were calculated on a weekly basis, by calculating the mean over one week for each animal.

## 5. Patents

S.H.E.K. and L.G. are co-holders of a patent on the tuberculosis vaccine candidate, VPM1002, licensed to Vakzine Projekt Management, Hannover, Germany and Serum Institute India Pvt. Ltd., Pune, India. 

## Figures and Tables

**Figure 1 ijms-24-05509-f001:**
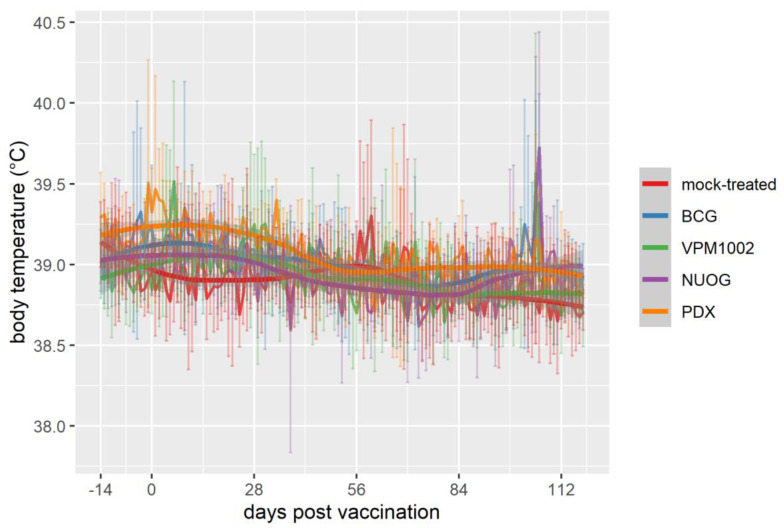
Body temperature of goats vaccinated with either *Bacille Calmette-Guérin* (BCG), VPM1002, VPM1002 ∆*nuoG* (NUOG), VPM1002 ∆*pdx1* (PDX), or mock-treated. Pictured are the mean temperatures (line plots in the background) and standard deviation (vertical error bars) per group, as well as the temporal development of temperatures smoothed by locally estimated scatterplot smoothing (LOESS) with 95% confidence bands (gray bands).

**Figure 2 ijms-24-05509-f002:**
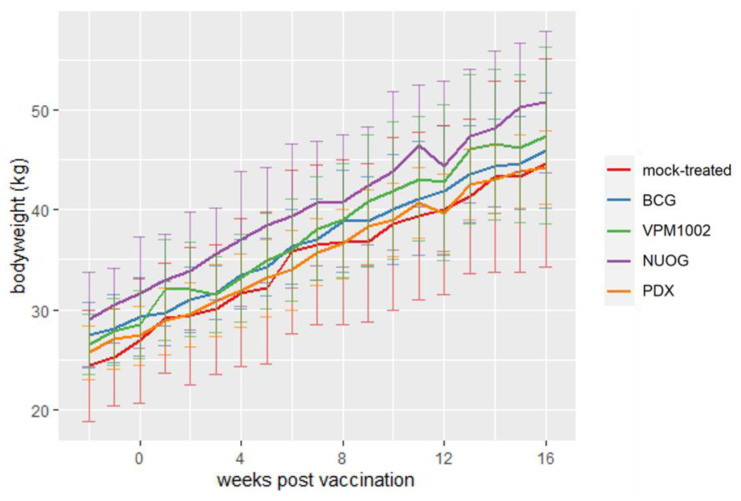
Changes in body weight of goats vaccinated with different vaccine strains (BCG, VPM1002, NUOG, PDX) and a mock-treated control group. Depicted is the mean body weight (line plots) and standard deviation (vertical error bars) per group.

**Figure 3 ijms-24-05509-f003:**
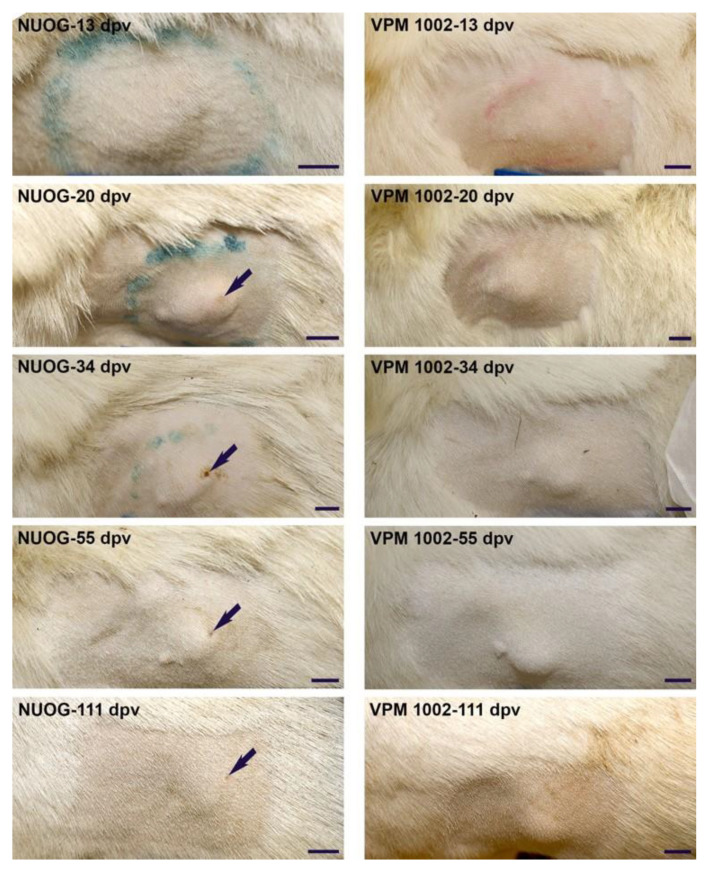
Representative photographs showing the development of skin lesions at the injection site from 13 to 111 days post vaccination (dpv). Pictured is a nodule with a superficial ulceration (arrow) of an NUOG-vaccinated goat (**left**) and a lesion covered by intact skin of a VPM1002-vaccinated goat (**right**). Size bar: 2 cm.

**Figure 4 ijms-24-05509-f004:**
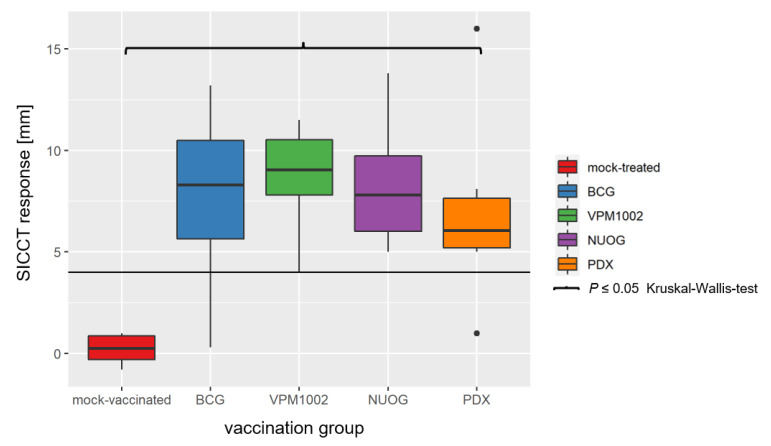
Results of the single intradermal comparative cervical test (SICCT) on 104 dpv of goats vaccinated with either BCG, VPM1002, NUOG, PDX, or mock-treated. Results show the change in the skin thickness after injection of bovine purified protein derivative (bPPD) minus the change in skin thickness after injection of avian PPD (aPPD; ΔbPPD–ΔaPPD). The horizontal line marks the limit for a positive test result (4 mm). All vaccinated groups statistically significantly differed from the mock-treated group.

**Figure 5 ijms-24-05509-f005:**
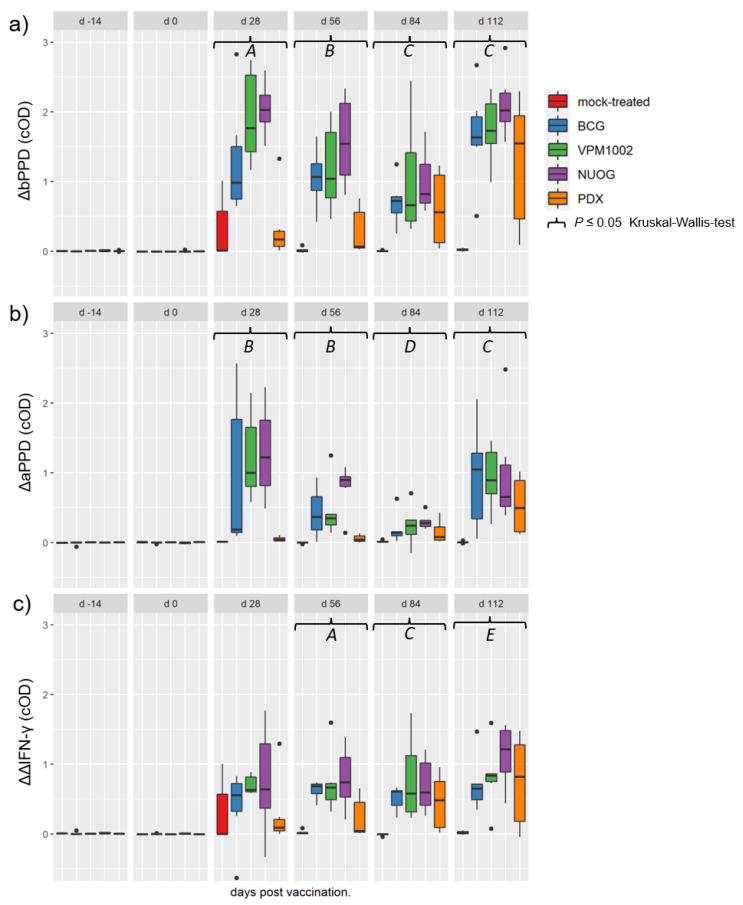
Antigen-specific interferon gamma (IFN-γ) release in whole blood samples of goats, subcutaneously vaccinated or mock-treated, upon in vitro stimulation with bovine (bPPD) and avian purified protein derivative (aPPD). Results are presented as Δ optical density of enzyme-linked immunosorbent assay (ELISA) measurements after stimulation of blood samples with bPPD (**a**), with aPPD (**b**), and as the difference between data obtained after stimulation with bPPD and aPPD (**c**), as described in Materials and Methods. (*A*) Statistically significant differences between the mock-treated group and the BCG-, VPM1002-, and NUOG-vaccinated groups, respectively. PDX-vaccinated animals significantly differed from the other three vaccinated groups, but not from the mock-treated group. (*B*) Statistically significant differences between the mock-treated group and all vaccinated groups, respectively, and between the BCG- and the NUOG-vaccinated group when compared to the PDX group. (*C*) Statistically significant differences between each of the vaccinated groups and the mock-treated group. (*D*) Statistically significant differences between the BCG-, NUOG-, and PDX-vaccinated, but not the VPM1002-vaccinated, groups and the mock-treated group. (*E*) Statistically significant differences between the BCG-, VPM1002-, and NUOG-vaccinated, but not PDX-vaccinated, animals and the mock-treated group.

**Figure 6 ijms-24-05509-f006:**
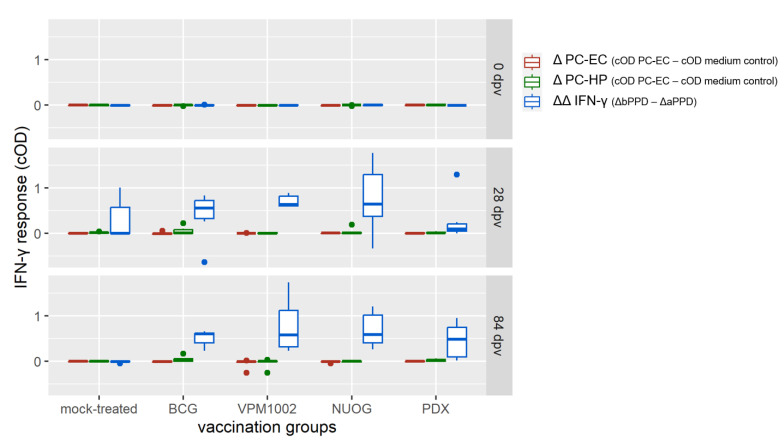
Interferon gamma (IFN-γ) release following stimulation with *Mycobacterium (M.) bovis*-specific peptide cocktails Bovigam^®^ PC-EC and PC-HP (Prionics Lelystad B.V.) of whole blood samples of BCG-, VPM1002-, NUOG-, or PDX-vaccinated and mock-treated goats. Data is presented as ΔIFN-γ values obtained after stimulation of blood samples taken at 84 dpv with PC-EC and PC-HP (cOD–cOD medium control) or with bovine and avian purified protein derivative (bPPD, aPPD, ΔΔIFN-γ), as described in Section 4.

**Figure 7 ijms-24-05509-f007:**
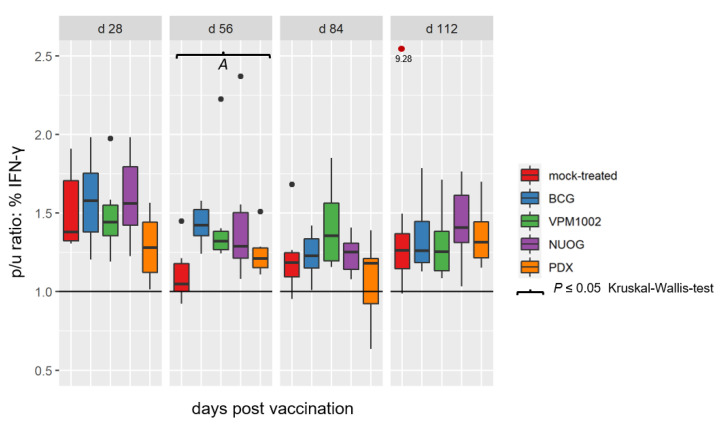
Relative proportion of total lymphocytes containing IFN-γ after in vitro stimulation with bPPD in peripheral blood mononuclear cells (PBMC) cultures derived from goats vaccinated with either BCG, VPM1002, NUOG, PDX or mock-treated. Results are expressed as p/u ratio $\left[\frac{\%\;bPPD\;stimulated\;cells\;\left(p\right)}{\%\;unstimulated\;cells\;\left(u\right)}\right]$ , as described in Materials and Methods. A p/u ratio greater than 1 (horizontal line) indicates an increased proportion of IFN-γ^+^ T cells after bPPD stimulation compared to unstimulated cells. (*A*) Cells from VPM1002- and BCG- vaccinated animals responded significantly differently to cells from the mock-treated goats.

**Figure 8 ijms-24-05509-f008:**
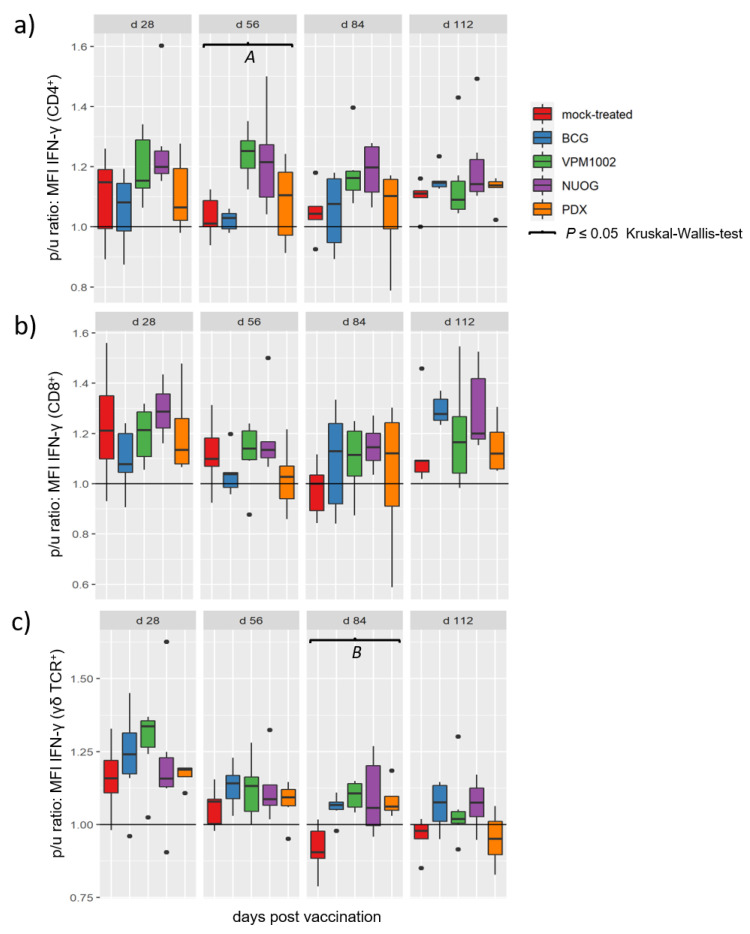
Relative amounts of intracellular IFN-γ in T cell subsets after in vitro stimulation with bPPD of goats vaccinated with either BCG, VPM1002, NUOG, PDX, or mock-treated. Boxplots represent values of the p/u ratio $\left(\frac{bPPD\;stimulated\;\left(p\right)}{unstimulated\;\left(u\right)}\right)$ of the median fluorescence intensity (MFI) for the detection of IFN-γ within the population of (**a**) CD4^+^ T cells, (**b**) CD8^+^ T cells, and (**c**) γδ T cells. (*A*) Values for the VPM1002- and NUOG-vaccinated groups significantly differed from the values of the mock- and the BCG-vaccinated groups. (*B*) Values of all vaccinated groups differed from values of the mock-treated group.

**Figure 9 ijms-24-05509-f009:**
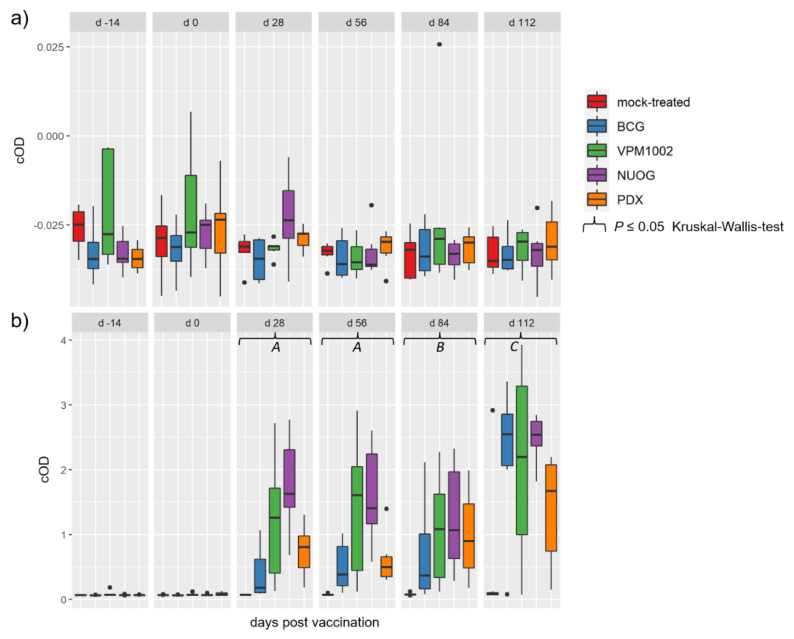
Humoral response to mycobacterial antigens of goats vaccinated with either BCG, VPM1002, NUOG, PDX, or the mock-treated control. (**a**) Antibodies recognizing *M. bovis* antigens (MPB70 and MPB83) and (**b**) *M. avium ssp. paratuberculosis* (MAP) antigens detected by respective ELISA tests (see Materials and Methods for details). (*A*) All vaccinated groups significantly differed from the mock-treated group. The MAP antibody response of the NUOG group significantly exceeded that of the BCG and PDX groups. (*B*) All vaccinated groups were statistically significantly different from the mock-treated group. (*C*) The BCG-, NUOG-, and PDX-vaccinated groups significantly differed from the mock-treated groups. The NUOG group response significantly exceeded that of the PDX group.

**Figure 10 ijms-24-05509-f010:**
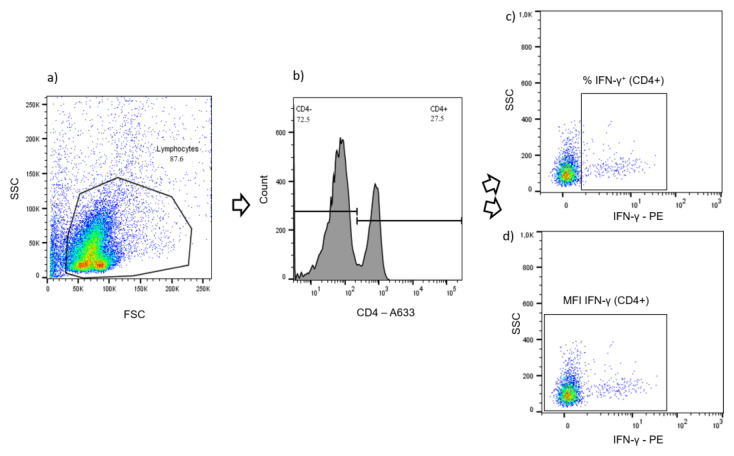
Flow cytometry gating strategy (representative example). (**a**) Gating for live and morphologically intact lymphocytes (density plot). (**b**) Gating for the CD4^+^ population (histogram). (**c**) Gating for the percentage of IFN-γ^+^ cells within the CD4^+^ population (density plot). (**d**) Gating configuration for the determination of the median fluorescence intensity (MFI) for the detection of IFN-γ in all CD4^+^ T cells (density plot).

**Table 1 ijms-24-05509-t001:** Skin lesions at the injection site of goats vaccinated with vaccine strains BCG, VPM1002, NUOG, PDX, or mock-treated. Mean (minimum; maximum) values of a total score per animal group (*n* = 6) are indicated. The total score describes the sum of the values for the size of lesion, redness, pain, swelling, local temperature, and necrosis (0 = no reaction, 1–4 = mild symptoms, 4–8 = moderate symptoms).

	Vaccination Group				
Weeks Post Vaccination (pv)	Mock-Treated	BCG	VPM1002	NUOG	PDX
−1	0.0 (0; 0)	0.0 (0; 0)	0.0 (0; 0)	0.0 (0; 0)	0.0 (0; 0)
1	0.0 (0; 0)	2.0 (1; 3)	2.2 (0; 4)	2.2 (0; 4)	2.2 (1; 3)
2	0.0 (0; 0)	3.2 (2; 5)	5.0 (3; 7)	5.7 (5; 6)	3.8 (3; 5)
3	0.0 (0; 0)	3.5 (2; 5)	5.5 (4; 7)	5.7 (5; 6)	3.5 (0; 6)
4	0.0 (0; 0)	3.7 (2; 5)	5.2 (4; 6)	5.8 (5; 7)	3.3 (0; 6)
5	0.0 (0; 0)	4.0 (3; 6)	5.3 (4; 6)	5.8 (5; 7)	3.2 (0; 5)
6	0.0 (0; 0)	4.0 (3; 6)	5.0 (3; 6)	6.0 (4; 8)	3.7 (2; 5)
7	0.0 (0; 0)	3.5 (2; 5)	4.5 (3; 5)	5.7 (4; 8)	3.3 (2; 5)
8	0.0 (0; 0)	3.2 (2; 5)	4.0 (3; 5)	5.2 (4; 7)	2.8 (2; 5)
9	0.0 (0; 0)	3.0 (2; 4)	4.2 (3; 5)	4.3 (4; 5)	2.7 (2; 4)
10	0.0 (0; 0)	3.0 (2; 5)	3.8 (3; 5)	4.5 (4; 5)	2.5 (2; 3)
11	0.0 (0; 0)	3.2 (2; 5)	4.2 (3; 5)	4.3 (4; 5)	2.7 (2; 4)
12	0.0 (0; 0)	3.2 (2; 5)	4.0 (3; 5)	3.5 (3; 4)	2.7 (2; 4)
13	0.0 (0; 0)	3.0 (2; 5)	3.7 (3; 5)	3.5 (3; 4)	2.8 (2; 4)
14	0.0 (0; 0)	3.0 (2; 5)	3.8 (3; 5)	3.3 (3; 4)	3.2 (2; 5)
15	0.0 (0; 0)	3.2 (2; 5)	3.5 (3; 5)	3.3 (3; 4)	3.0 (2; 5)
16	0.0 (0; 0)	3.2 (2; 5)	3.5 (3; 5)	3.5 (3; 4)	2.7 (1; 5)
17	0.0 (0; 0)	3.0 (2; 5)	3.5 (3; 5)	3.7 (3; 5)	2.3 (0; 5)
18	0.0 (0; 0)	3.0 (2; 5)	3.2 (2; 5)	3.5 (3; 4)	1.8 (0; 4)

**Table 2 ijms-24-05509-t002:** Detection of granulomas, re-isolation of vaccine strains, and detection of acid-fast bacteria (AFB) by Ziehl–Neelsen staining from goats vaccinated with either BCG, VPM1002, NUOG, PDX, or mock-treated. Tissues, including the site of vaccination, *Lymphonodus (Ln.) cervicalis superficialis sinister (sin.)*, *Ln. axillaris profundus sin.*, and *Lymphonodi (Lnn.) mediastinales*, were obtained at necropsy 127 dpv.

	Number of Positive Goats/Number of Goats Per Group													
Group	Vaccination Site				*Ln. cervicalis superficialis sin.*				*Ln. axillaris profundus sin.*				*Lnn. mediastinales*	
	Granuloma	AFB	Cultivation of Vaccine Strains		Granuloma	AFB	Cultivation of Vaccine Strains		Granuloma	AFB	Cultivation of Vaccine Strains		Granuloma/AFB	Cultivation of Vaccine Strains
mock-treated	0/6	0/6	0/6		0/6	0/6	0/6		0/6	0/6	0/6		0/6	0/6
BCG	6/6	6/6	3/6		1/6	1/6	0/6		1/6	1/6	0/6		0/6	1/6
VPM1002	6/6	6/6	2/6		1/6	1/6	0/6		1/6	1/6	0/6		0/6	0/6
NUOG	6/6	5/6	1/6		0/6	0/6	0/6		0/6	0/6	0/6		0/6	0/6
PDX	5/6	4/6	0/6		2/6	2/6	0/6		1/6	0/6	0/6		0/6	0/6

**Table 3 ijms-24-05509-t003:** Scoring system of local lesions at the injection sites.

Symptom	Description	Score
size of the lesion	no lesion	0
	<2 cm diameter	1
	2–5 cm diameter	2
	5–10 cm diameter	3
	>10 cm diameter	4
edematous swelling	no	0
	yes	1
color	physiological	0
	redness, other color	1
texture	physiological	0
	alteration	1
pain	no	0
	yes	1
local temperature	normal	0
	increased	1
necrosis	no	0
	yes	5
suppuration	no	0
	yes	5
total score	no reaction	0
	mild reaction	1–4
	medium reaction	4–8
	strong reaction	9–14

## Data Availability

The data presented in this study are available on request from the corresponding author.

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
