# Peer review of "Safety and Immunogenicity of Recombinant Bacille Calmette-Guérin Strain VPM1002 and Its Derivatives in a Goat Model"

_ijms, 2023, doi:10.3390/ijms24065509_

Round 1
Reviewer 1 Report (Previous Reviewer 1)
Your position regarding my comments are appropriate, it would seem to be the basis for a future paper, well done.
Author Response
We sincerely thank the reviewer for her/his comments and appreciate the encouragement to further studies on the topic.
Reviewer 2 Report (Previous Reviewer 2)
This is an well written and presented manuscript describing immumological responses of goats to BCG and BCG-derived vaccines for tuberculosis. I was appreciative to see such vaccines evaluated in goats instead of simply another mouse model. I do not have any substantive criticisms - the work was comprehensive and multifaceted, and it appeared to be executed expertly. One suggestion on an topic that I, and I suspect, other readers would like to see expanded somewhat if possible is the idea of correlates of protection for in vitro testing relative to TB vaccines. You mention this briefly on line 453, but as a reviewer not well versed in the TB literature, it would be great to expand this discussion to answer the question: is there anything short of challenge experiments to evaluate whether the derivative vaccines evaluated here might indeed be superior to BCG itself. I suspect the answer is no and it appears that challenge experiments are planned.
Author Response
We thank the reviewer for this very important suggestion. We got to emphasize that there are no reliable immune correlates of protection (CoP) for TB vaccines defined yet, neither in challenge models nor in clinical human or animal contexts. The efficacy of the novel vaccine candidates cannot be predicted by the immunogenicity data we presented.
We highlighted this fact in our conclusions (see line 548-553), but refrained from further elaborating on CoP against TB in the recent manuscript. These data were derived from the first experiment in a series of trials. Indeed, we subsequently established a caprine TB challenge model and will present efficacy data for VPM1002 and its derivatives in goats. This will allow a more extensive discussion of the assessed immune parameters in regard to protection provided by the vaccine candidates in goats.
As was summarized, e.g. by Fletcher et al. (2018) or Bhatt et al. (2015), the definition of CoP for TB vaccines remains a challenging task and requires the successful completion of phase III efficacy trials. Nevertheless, large animal TB models like cattle and goats might be a valuable contribution to identifying potential CoP, as challenge experiments in more natural models than mice can be conducted under standardized conditions in a relatively short period of time as compared to human clinical trials.
References:
Fletcher, H. A. (2018). "Systems approaches to correlates of protection and progression to TB disease." Semin Immunol 39: 81-87.
Bhatt, K., et al. (2015). "Quest for correlates of protection against tuberculosis." Clin Vaccine Immunol 22(3): 258-266.
Reviewer 3 Report (Previous Reviewer 4)
The manuscript by Figl et.al describes the immunological profiling of BCG, VPM1002, VPM1002-PDEX, and VPM1002-NUOG in juvenile goats. The authors measured various parameters as surrogates of immunogenicity induced by these vaccines/candidates. The article is easy to follow and written with clarity from the introduction to the conclusion sections. Although this work is important and the paper is worth considering further, there are 3 major issues that need to be addressed through major and explicit revision/edits.
1). The same group of authors published a similar article (PMID: 36232295) in the same journal in 2022 (cited as ref.31) in the current paper. Ironically, in that published paper, the authors cited the current manuscript as “submitted” (cited as ref.27 in the last year’s publication). It is unfortunate and concerning that the authors pre-determined these two articles as “part-1” and “part-2 (submitted)”, and cited “part-2” before even it is published !. Another error/mistake is the pubmed/google scholar link provided to ref.27 in the “part-1” paper, which opens to the main/parent article. The authors should make efforts to fix these errors. Apart from these confusions, it is important that the authors must disclose if the animals used and the samples collected for both papers (i.e, PMID: 36232295 and the current paper) originated from the same cohort or different cohort of animals (i.e., n=30 per each study separately).
2). Despite two papers (one published and one submitted) on the safety and immunogenicity in goats, no data on the protective efficacy of these tested vaccines against pathogenic mycobacterial (e.g M. caprae) infection was provided. This is a major weakness of this study. I assume that the authors are very well aware that the impact, usefulness, and significance of the immunogenicity elicited by a vaccine can only be validated through its ability to protect (or not) the vaccinated host. Considering the reputation of this research group (particularly Dr. Kaufmann), it is surprising to see more immune characterization without efficacy testing in this model (refer to a PLoS ONE paper published in 2013; PMID: 24278420).
3). There are several confounding factors that might have interfered with the immune parameters and clinical readouts reported in this paper. These factors include, (1), the juvenile nature (as opposed to adult animals); 2). castration of the animal (vs. non-castrated); 3). Vaccination with Heptavac P plus; 4). Presence of various infectious agents, including Staph. aureus and Moraxella bovis in several animals in the study; 5). Treatment with antibiotics before testing the vaccines; and 6). a positive antibody titer against MAP. The authors should at least acknowledge the effect of these factors (except MAP) on the data obtained and interpreted in the discussion section.
On a minor note, the language should be improved for scientific soundness. For example, avoid writing “our animals” (line 507, 522), “our goats” (line 520, 493, 478).
Round 2
Reviewer 3 Report (Previous Reviewer 4)
This reviewer appreciates the efforts taken by the authors to address my comments and revise the manuscript.
Although most of the authors' responses are satisfactory, the issue of animal usage still needs an explicit declaration. Therefore, I suggest the authors mention whatever they responded to in point#1 of their "reply to the reviewer" document.
Specifically, the following should be included in Section 4.1. Experimental animals, housing and health status:
"The data presented in a prior publication by our group and the current manuscript are based on samples obtained from the same cohort of animals. We consider this an important implementation of the 3R principles (Replacement, Reduction, and Refinement) for the use of experimental animals. However, there is a clear distinction between the experimental approaches used in these papers: While in this manuscript results pertaining to systemic immunity and safety are reported, our previous paper by LieblerTenorio et al. focuses on the detailed morphologic analysis of the vaccine granulomas."
I believe this information is important for ethical and transparency reasons and will help in a better understanding of the overall research projects conducted by this group.
Author Response
We thank the reviewer for her/his comment and revised the manuscript accordingly by adding the suggested passage to the respective section.
This manuscript is a resubmission of an earlier submission. The following is a list of the peer review reports and author responses from that submission.
Round 1
Reviewer 1 Report
This article articulates the need to improve BCG efficacy and safety. The authors tested a recombinant BCG strain, VPM1002 and its derivatives, employing them and placebo in a goat animal model. The NUOG derivative induced a robust cellular immune response and an acceptable safety profile.
Strengths:
Selecting BCG strain VPM1002 for is phagosome-modifying virulence factor and then modifying it by deleting ureC for increased cytosol antigen load.
Generating safety (PDX) and efficacy (NUOG) BCG-VPM1002 variants.
Selection of the caprine TB model.
Minor recommendations:
In the introduction and/or discussion section, the article could be strengthened by addressing newfound “off-target” use of BCG in autoimmune and neurodegenerative diseases and how development of alternatives to current strains of BCG should be evaluated in that light:
Angelidou A, Pittet LF, Faustman D, Curtis N, Levy O. BCG vaccine's off-target effects on allergic, inflammatory, and autoimmune diseases: Worth another shot? J Allergy Clin Immunol. 2022 Jan;149(1):51-54. doi: 10.1016/j.jaci.2021.09.034. Epub 2021 Oct 18. PMID: 34673049.
and
Singh AK, Netea MG, Bishai WR. BCG turns 100: its nontraditional uses against viruses, cancer, and immunologic diseases. J Clin Invest. 2021 Jun 1;131(11):e148291. doi: 10.1172/JCI148291. PMID: 34060492; PMCID: PMC8159679.
Reviewer 2 Report
Control of tuberculosis in humans and animals is clearly a high priority and the studies reported here extend out understanding of the efficacy and safety of bCG derivatives that may offer enhanced immunization capacity relative to standard bCG. I consider this a well designed and executed study to assess safety and immunogenicity of 3 bCG derivatives using a goat model. While the results did not point to these new strains being "game changers", there were indications that they were superior in some ways to traditional bCG. I do not have significant criticisms of the presentation or content and judge this work to be a valuable contribution to the ongoing quest for improved TB vaccines.
Reviewer 3 Report
1. The overall contents of the manuscript does not seem to fit the topic of this journal.
2. The description of the molecular mechanisms of action of the tuberculosis vaccine proposed by the authors is insufficient.
3. The discussion part is too simplistic and has a lot of volume.
Reviewer 4 Report
This manuscript claims to report the clinical, pathological and immunological aspects of vaccination with 4 vaccine candidates (BCG, PDX, NUOG and VPM1002) in a goat model. The manuscript is written in a standard language and flow. Standard reagents and tools were used for various experiments. Although it appeared as an important study at the outset, the inherent issues with the nature of animals used in this study outweighs the merits of the findings. The apparent lack of well-established and anticipated immunological features of typical BCG-based vaccines in this goat model appears to be inconsistent and non-reproducible with the responses of the similar vaccines tested in the non-human primates and/or in humans, which diminishes the enthusiasm of this article. Some of the major concerns are as follows:
RESULTS:
1). A major concern is the lack of histological images of various pathological manifestations and AFB in the lesions, which are elaborated in the text. Similarly, no histological evidence was provided for the claims in lines 226-228.
2). Line 211-214. Define the different types of granulomas (type1, 2 and 3) and provide representative histological images.
3). The observation in lines 229-231 could be contributed by the confounding, underlying clinical signs that was noted in animals prior to vaccination (refer to lines 639-640 of the methods section).
4). What is the justification for the finding of other mycobacterial species (M. septicum / M. peregrinum and M. gordonae) that were different from the one used for vaccination in the goat tissue samples? (Lines 233-234). This observation suggests a potential interference of these mycobacterial species on the clinical, histopathological and immunological findings reported for different vaccines in this study.
5). Figure-2. Although it is mentioned that all animals were at an age of 5 months, the starting bodyweight was strikingly different (more than 10kg ?!) for various vaccine groups, for example, an animal in the PDX group got less than 17kg, while an animal in the NUOG group got about 34 kg, which is almost twice the weight of the former group. This needs to be explained.
6). Figure-3. The representative image of skin lesions of mock (negative control), BCG and PDX groups are missing. These are important groups that would validate the observations of VPM1002 and NUOG groups shown in this figure; therefore, the photographs of missing group should be added or this figure-3 should be removed.
7). Figure-5 and 6. It is very interesting to note that the mock-vaccinated animals had a positive value for bPPD and IFN-g only on 28d, although it didn’t show any positive response with PC-EC or PC-HP at that time. What is the cut-off or limit of detection of the assays used in figures 5 and 6 ? (it looks like for IFN its 0.1),
8). Figures-7 and 8. It is interesting to note the IFN-g response from all 3 T cell subsets of the mock-treated animals. The extent of the response in this group seems comparable (even better than BCG in Fig 8b) to all other groups at most of the tested timepoints. What could be the reason? Were the mock-vaccinated animals cross-contaminated with other groups or infected with NTMs ?.
9). Figure-8, it is very disturbing to note that antibodies to MAP antigens were measured, while no antibody was detected for BCG-antigens (line 338-342). I am afraid if this observation suggests a major and inherent flaw that is fundamental to the animals used for this study.
10). Figure-10. The MFI and % IFN+ CD4+ gating is confusing. Why the panel d (non-selective panel) was included, when the % of IFN+/CD4+ cells can be seen in panel-c (more selective) ?
11). Table-1, it is stated that “Mean (minimum; maximum) values of a total score per animal group 163 (n=6) are indicated”. However, several values appear incorrect. Example, the BCG group at 3 weeks shows that the minimum was 2, and the maximum was 5, and the mean was 2. How is this possible?. Even if one assumes that 5/6 animals had a score of 2 (minimum), and 1/6 animal had a score of 5 (maximum), then the mean should be ((5x2)+ (1x5))/6 = 2.5 (n=6) and not 2. Please check your calculations or explain.
DISCUSSION:
1). Inconsistency in the vaccine dose between methods and discussion sections (compare the lines 660-663 with line 474-476).
2). The argument related to dose of vaccine versus immune response (line 482-485) is speculative; no evidence is shown for this effect. What is the rationale that goats’ response won’t be like the deers’ response in the study cited in ref#52 and that the assays used were not conclusive/appropriate for this effect ?
3). It appears that most of the justifications for the lack of clear immunogenicity of tested vaccines in this study were speculative, mostly citing circumstantial literature support from mouse, human and cattle studies, which is not convincing for the negative/differential observation presented.
METHODS:
1). Mention the name of the vendor that supplied the goats
2). Goats have problems with presence of various bacteria (Mannheimia haemolytica, Moraxella bovis and Staphylococcus aureus), animals were treated with two different drugs (Enrofloxacin, Toltrazuril) showed pathologic clinical signs such as chronic bronchopneumonia, chronic bronchitis or pleuritis.
3). BCG-SSI was administered at a much lower dose (almost 10x less than NUOG vaccine), compared to the other candidate vaccines.
4). What is the rationale for the scoring scale of necrosis and suppuration, and for total score (cite any reference, if applicable).
Line 668. How was the painfulness of lymph nodes measured ?
Line 681. Why was vit.B6 measured ?
Line 684. Check if it is Leuko-TIK or Leuko-TIC ?.
Line 697. Mention the names of the 4 “other mycobacterial genes” in Bovigam PC-HP.
Line 714, Check if it is Puffer or Buffer ?.
Line 721. Why was the PBMCs (stimulated with bPPD or conA or none) incubated with PMA plus Ionomycin ?
Line 728. It is unclear if the antibodies used were monoclonal or polyclonal. This information must be provided.